# A Feasibility Study on Utilizing Remote Sensing Data to Monitor Grape Yield and Berry Composition for Selective Harvesting

**DOI:** 10.3390/plants14010088

**Published:** 2024-12-31

**Authors:** Leeko Lee, Andrew Reynolds, Briann Dorin, Adam Shemrock

**Affiliations:** 1Department of Biological Sciences, Brock University, 1812 Sir Isaac Brock Way, St. Catharines, ON L2S 3A1, Canada; andrewreynolds2007@gmail.com (A.R.);; 2AirTech UAV Solutions Inc., Inverary, ON K0H 1X0, Canada; ashemrock@airtech-solutions.ca

**Keywords:** remotely piloted aircraft system (RPAS), precision viticulture, vegetation index (VI), NDVI, thermal emission, remote sensing, grapes, yield, phenols, anthocyanins

## Abstract

The primary purpose of this study was to improve our understanding of remote sensing technologies and their potential application in vineyards to monitor yields and fruit composition, which could then be used for selective harvesting and winemaking. For yield and berry composition data collection, representative vines from the vineyard block were selected and geolocated, and the same vines were surveyed for remote sensing data collection by the multispectral and thermal sensors in the RPAS in 2015 and 2016. The spectral reflectance data were further analyzed for vegetation indices to evaluate the correlation between the variables. Moran’s global index and map analysis were used to determine spatial clustering patterns and correlations between variables. The results of this study indicated that remote sensing data in the form of vegetation indices from the RPAS were positively correlated with yield and berry weight across sites and years. There was a positive correlation between the thermal emission and berry pH, berry phenols, and anthocyanins in certain sites and years. Overall, remote sensing technology has the potential to monitor and predict grape quality and yield, but further research on the efficacy of this data is needed for selective harvesting and winemaking.

## 1. Introduction

A significant increase in research on the application of precision viticulture (PV) for vineyard management has occurred over the last few decades. Spatial maps of soil variability and airborne images of vineyards have demonstrated the potential for zonal management at the subfield level [1,2,3]. Zonal management in vineyards is the management of subfield or sub-block regions that differ in traits, such as vine physiology, yield, and fruit quality [4,5,6]. The zonal or selective harvesting of grapes has the potential to increase profits for grape growers [7,8]. Furthermore, PV has also brought many environmental benefits due to the more efficient and targeted use of farm resources, such as fertilizer and pesticides [9,10]. Even though PV research has highlighted the potential benefits of zonal management, many grape growers tend to perform uniform management in a single vineyard block, despite the fact that various environmental and biological factors can have effects on fruit composition and the productivity of grapevines within a single block [11,12].

Previous research on PV has mainly focused on defining methods to identify the spatial variation within a single vineyard block such that, if the spatial variation can be identified, the vineyard can be managed differently in these subfield regions to maximize the yield and/or quality [13,14]. Research has demonstrated significant vineyard variability in fruit composition and the productivity of grapes due to differences in vine physiology or environmental variables (e.g., canopy size, sun exposure, water status) within individual vineyard blocks [1,15,16,17,18,19,20,21,22,23,24,25,26,27]. For example, vineyard soils can vary within a block, in accordance with the soil composition and nutrient levels [28,29,30]. These variabilities have been shown to impact grape composition and quantity in single vineyard blocks of Riesling [31] and Cabernet franc [32].

Direct in situ sampling for the yield and berry composition is a common method of delineating management zones, despite several challenges and limitations [33], which is why the use of remote sensing technology has become increasingly common to establish zonal vineyard management. Technical developments have made it possible to collect remote sensing data closer to the area being studied with greater spatial resolution. These advancements have also resulted in easier access to data and are more cost-effective [33,34]. As an alternative to traditional methods, using remotely piloted aircraft systems (RPASs), the rapid sampling of large areas is possible [35]. RPAS technology flies at much lower altitudes, enabling the collection of imagery at much higher spatial resolutions, in the range of 1 cm2 pixel size [36]. This type of remote sensing platform is gaining scientific interest, including its use in precision viticulture.

Several promising approaches to mapping vineyard leaf water potential [37,38] and vegetative growth [39] have been suggested. The electromagnetic reflectance of vegetation in agriculture is measured in multiple wavelength bands, mostly the green, red, red edge, and near-infrared (NIR) bands. The factors contributing to the different level of reflectance in green, red, red edge, and NIR peaks are very diverse. Leaf reflectance in green and red is usually controlled by plant pigments, while, in the NIR range, variations in reflectance might be influenced by alterations in the leaf structure and/or thickness that affect leaf absorbance and reflectance [40]. The red edge peak is in the boundary between visible and NIR spectrums, and the peak is studied to determine foliar composition, such as chlorophyll content [41]. Chlorophyll is the primary molecule that absorbs light energy and converts the energy for photosynthesis and is also a critical barometer of plant health and growth [42]. Chlorophyll content is negatively correlated to the red reflectance peak, because chlorophyll strongly absorbs red radiation [42]. Therefore, red reflectance could be a reliable candidate for estimating plant chlorophyll content and photosynthesis activity. However, the measurements of red reflectance are very sensitive to the effects of various other variables, such as solar irradiance, the presence of other pigment molecules, and background soil [40,43]. Reflectance at the red edge is less influenced by these variables, since it marks the boundary between the inner leaf cellular scattering in the NIR peak and chlorophyll absorption in the red peak [44]. Numerous vegetation indices from remote sensing data can be calculated from the green, red, red edge, and NIR regions of EM reflectance [45]. The normalized difference vegetation index (NDVI), for example, was calculated by converting reflectance data into a ratio of near infrared to red reflectance [46]: NDVI = [(near infrared)-(red)]/[(near infrared) + (red)]. Other remote sensing indices in viticulture are also well studied [47], including indices such as green, red, red edge, NIR, the green chlorophyll index (CI green), the red edge chlorophyll index (CI red edge), the red edge normalized difference vegetation index (NDRE), the NDVI green (GNDVI), and the ratio vegetation index (RVI) [48,49,50,51]. Sensors for measuring the thermal energy emitted in the infrared region by plants have also been used in PV, primarily for detecting water stress caused by an increase in the leaf temperature [52]. Through remote sensing software, vegetation indices can be used to map out unique vineyard zones, which can then be linked to grapevine yield and fruit composition [53,54,55,56,57,58].

Despite substantial research undertaken to investigate the use of remote sensing data to detect vineyard variation across a range of grapevine and crop variables, the effectiveness of these technologies in optimizing vineyard management and their potential use for selective harvesting needs further study. Our study aims to improve our current knowledge about remote sensing applications in precision viticulture to improve monitoring the grape yield and fruit composition. We hypothesized that vegetation indices calculated from remote sensing data from RPAS flights would correlate with manually measured variables of grape yield and berry composition. Maps of these variables could indicate the usefulness of this technology for the zonal or selective harvesting of grapes based on remote sensing data. This research provided an in-depth insight into how remote sensing technologies can enhance vineyard production monitoring through an exploration of a wide range of grape yield and fruit composition variables and their relationship to remote sensing data.

## 2. Results

### 2.1. Principal Component Analysis (PCA) of Remote Sensing Data and Grape Yield/Fruit Composition

PCA results were built based on the first two factors, which explained between 44 to 58% of the data (Figure 1 and Figure 2). A positive association was found between the NDVI and yield at three out of five sites in 2015 and at four out of six sites in 2016. The NDVI and berry weight were clustered together at four out of five sites in 2015 and at four out of six sites in 2016. There was no clustering between the NDVI and the number of clusters in 2015, and only three sites showed clustering between the two variables in 2016. Only one site showed a positive association between the NDVI and the level of phenols in 2015 and between the NDVI and TA in 2016. Thermal emission data and pH were clustered together on only one site in 2015 and all six sites in 2016. Thermal data were also associated with Brix at only one site in 2015 and at four sites in 2016. The same clustering pattern was found between the thermal data and phenols/anthocyanins. However, many berry composition data were derived from relatively short vectors, so visual correlations were difficult to establish. Pearson’s correlations were used to determine direct relationships between variables.

### 2.2. Pearson’s Correlation Between Analysis of EM Spectra Data and Grape Yield/Fruit Composition

Table 1 indicated that a positive correlation was found between the NDVI and yield at two out of five sites in 2015 and five out of six sites in 2016, as well as berry weight at five sites in 2015 and three sites in 2016. However, there were only weak correlations observed between thermal emission and yield with negative relationships at three sites in 2016. Significant positive correlations between the NDVI and yield were observed in five of six sites in 2016, while only two vineyards had the correlation in 2015. There were only weak correlations found between the NDVI and anthocyanins/phenols and a lack of consistency in the correlations was evident throughout the sites and years. The NDVI was only negatively correlated to anthocyanins/phenols at one site each year. Thermal imaging was positively correlated with anthocyanins at two sites and with phenols at three sites out of six sites in 2016; whereas, only one site showed the correlation in 2015. The NDVI had a low capability of detecting variation in anthocyanins and phenols. There were only a few sites showing correlations between the NDVI/thermal and Brix/TA and a lack of consistency in the correlations was evident throughout the sites and years. However, thermal imaging data showed a strong positive correlation to the pH at all six sites in 2016 and site 5 also showed temporal stability.

### 2.3. Analysis of Correlation Between Other Indices from the RPAS Flight and Grape Yield/Fruit Composition

Since there was a lack of correlation between the NDVI and thermal data in 2015, this study examined other vegetation indices (VIs) in 2016 data to investigate the feasibility of remote sensing technologies to monitor the variations of plant yield and fruit quality.

#### 2.3.1. Principal Component Analysis (PCA) of Other Indices from the RPAS Flight and Grape Yield/Fruit Composition

PCA results were presented in Figure 3, and the first two factors explained between 50 to 62% of the data.

In general, the yield indicated a positive association with the CI green, GNDVI, and RVI, while these were oppositely associated with green and red. It was observed that berry composition data did not group well with other indices from the RPAS flight. In sites 2 and 3, pH and phenols were positively associated with the green, red, red edge, and NIR. A positive association was also observed between anthocyanins and green/red edge/NIR in site 2, and the same association was found between anthocyanins and green/red in site 3. The negative associations observed between anthocyanins and green/red edge/NIR in site 2, and the same association was found between anthocyanins and green/red in site 3. However, pH and phenols were inversely correlated to green, red, red edge, and NIR in site 5. As shown in their short vectors, berry weight, Brix, pH, and TA could not be well explained by the first two factors.

#### 2.3.2. Pearson’s Correlation Analysis of Indices from RPAS Flight and Grape Yield/Fruit Composition

##### Relationships Between Other Indices and Yield Components

In Table 2, CI green was positively correlated to yield at five sites. Similar to CI green, strong positive correlations were observed between the GNDVI and yield at five out of six sites. RVI also had a positive relationship with berry weight in four sites, and with yield in three sites. In sites 3 and 4, positive correlations were observed between the CI green/GNDVI/RVI and all three yield components.

##### Relationships Between Other Indices from the RPAS Flight and Berry Composition

There were some correlations observed between other indices from the RPAS flight and berry composition data. Three sites (site 2, 3, and 5) had some correlations between the two variables (Table 2). The CI green, CI red edge, GNDVI, and NDRE were negatively correlated to Brix, pH, TA, phenols, and anthocyanins at one or two out of six sites, while the nature of the relationships in the other indices varied between sites. The correlations between indices from the RPAS flight and berry compositions were site-specific, as two sites (sites 2 and 3) showed similar correlations between the two variables, while an inverse correlation observed in another site (site 5). The other three sites only indicated weak correlations between the two variables, which varied between the sites. Interestingly, in site 3, CI green, green, and red were positively correlated with all berry composition data; whereas, those were adversely correlated with all yield component data.

### 2.4. Mapping and Spatial Autocorrelation Analysis

The spatial autocorrelation of each variable was determined by Moran’s index (Table 3), which determines clustering patterns and can indicate the feasibility of zonal vineyard management options [1,59,60,61]. The NDVI and thermal imaging were highly clustered across the six sites throughout the years. The cluster numbers in 2015 and 2016 showed a generally random distribution, being clustered at only two sites in each year, and a dispersed pattern also appeared at site 4 in 2016. The yield was mostly clustered in 2016 with five out of six sites being clustered; only site 1 was randomly distributed. However, there was no clustering pattern observed in 2015, and the yield was randomly distributed in all the vineyards. There was a strong clustering of anthocyanins across sites, and the pH level showed clustering patterns in 4 out of 6 sites in 2016 (hot and dry year). Overall, pH and anthocyanin levels showed high clustering; whereas, the number of clusters and yield were predominantly random. It is imperative that reliable maps are produced that show the areas of substantial variation to determine if data from the RPAS flight will be useful in detecting vineyard variability.

Figure 4, Figure 5, Figure 6, Figure 7, Figure 8 and Figure 9 show the site 1 to 6 maps of remote sensing data analysis from RAPS flight, yield components, and berry compositions in 2015 and 2016. In site 1 (Figure 4), the NDVI maps showed low values in the southwestern zone of the block and high values in the northeastern zone. The thermal maps showed an inverse correlation with the NDVI maps. Yield and clusters indicated an inverse spatial pattern to the NDVI in 2015, while a positive correlation to the NDVI in 2016. TA had an inverse pattern to the NDVI in both years, with high values in the southwestern zone of the block. Most of the clustering occurred in the east–west spatial variation for the berry composition data, while the NDVI data indicated north–south clustering. Therefore, the spatial correlation between the remote sensing data and berry compositions was weak and limited. Red edge and NIR maps indicated more east–west spatial clustering with a high value in the western block, which showed inverse patterns to TA and phenols.

In site 2 (Figure 5), the NDVI maps showed a lower NDVI in the southwestern and northeastern portion of the block and a higher NDVI in the northwestern zone of the block. Those also showed highly patchy spatial patterns with high and low values seen throughout the entire site. The thermal maps showed an inverse pattern to the NDVI maps in 2015, but the relationships were unclear in 2016. The yield components indicated similar spatial patterns to the NDVI in both years. Clusters and yield with spatial clustering of the low southwestern block and high northeastern showed an inverse pattern to green, red edge, and NIR. The yield also showed a similar spatial pattern to the CI green, CI red edge, NDRE, and GNDVI. Brix, pH, phenols, and anthocyanins were negatively correlated with the NDVI in both years, with high values in the northeastern zone of the block, while TA showed an inverse correlation with the NDVI. Green, red edge, and NIR indicated a clear spatial clustering in the low southwestern block and high northeastern block, which showed similar spatial patterns to brix, pH, phenols, and anthocyanins. These variables also showed an inverse pattern to the CI red edge and NDRE in the high southwestern block.

In site 3 (Figure 6), the NDVI map showed similar spatial patterns to those of the clusters, yield, and berry weight, with lower values in the north end and the center of the block and with a finger shape and higher values in the south end and north side of the center. The yield components had a similar spatial cluster pattern to the CI green, CI red edge, NDRE, GNDVI, and RVI, with lower values in the north end and the center of the block with a finger shape, while the yield and berry weight showed an inverse pattern with green, red edge, and NIR. An inverse spatial pattern with the NDVI maps was also seen in the maps of brix, pH, TA, phenols, and anthocyanins, where lower values were seen in the south end and the north side in the center of the map and higher values in the northwestern and central eastern regions. Berry compositions had a similar spatial cluster pattern to green, red edge, and NIR, with higher values in the north end and the center of the block with a finger shape, while the brix, pH, and phenols showed inverse patterns with CI green, CI red edge, NDRE, GNDVI, and RVI.

In site 4 (Figure 7), the NDVI maps displayed an unexpected and unusual diagonally striped pattern. The map showed a low NDVI along the northwestern edge of the block and a higher NDVI in the southeastern portion of the block and showed similar spatial patterns. The thermal maps only showed inverse patterns to the NDVI maps in 2016, while it displayed two unusual big circle patterns in the middle of the site in 2015. The yield and berry weight indicated a similar spatial pattern to the NDVIs through the years. The yield components had a similar spatial cluster pattern to red edge, NIR, CI green, CI red edge, NDRE, GNDVI, and RVI, with higher values in the south end and the northeastern of the block, while the yield showed inverse patterns to green and red with a north–south striped pattern. Inverse spatial patterns to the NDVI maps were also seen in the maps of anthocyanins, where lower values were seen in the south end of the map. The anthocyanins had an inverse spatial cluster pattern to red edge, NIR, CI green, GNDVI, and RVI, with higher values in the south end of the block, while brix had a similar clustering pattern to green and red.

In site 5 (Figure 8), the yearly NDVI maps showed similar spatial patterns with a higher NDVI in the eastern side of the block and along the southern edge of the block and lower values in the central and northeastern areas of the block. Thermal maps showed inverse patterns to the NDVI maps in both years. The yield and berry weight indicated similar spatial patterns to the NDVIs through the years. Clusters and yield had a similar spatial cluster pattern to red edge and NIR, with higher values in the southwestern side of the block. Similar spatial patterns to the thermal maps were seen in the maps of brix, pH, phenols, and anthocyanins, where lower values were seen in the southwestern side of the map. The maps of brix, pH, and phenols had an inverse spatial cluster pattern to green, red, red edge, and NIR, with higher values in the southwestern side of the block.

In site 6 (Figure 9), the NDVI maps showed a low NDVI along the central eastern and edge of the south block and higher values in the other south portion of the block. The thermal maps showed inverse patterns to the NDVI maps through the years. The yield and berry weight indicated a similar spatial pattern to the NDVIs over the years, while these correlations were weaker in the maps of thermal data. The yield and clusters had a similar spatial cluster pattern to red, with higher values in the west end of the block, while the berry weight showed an inverse pattern to red.

Overall, the NDVI maps indicated similar clustering patterns to anthocyanins in most sites. The thermal emission maps showed less similarity in spatial patterns to the yield component maps, but these indicated more similarity in spatial patterns to the berry composition maps. Especially, the thermal emission maps showed similar spatial patterns identical to the pH at all six sites in 2016. CI green and GNDVI maps showed the most similar spatial patterns to yield, and similar patterns were also found between the RVI and berry weight in four of six sites.

## 3. Discussion

This study demonstrated the viability of using remote sensing data for determining vineyard productivity and fruit quality in cool-climate Cabernet franc vineyards. Consistent relationships were seen between the NDVI and measures of total yield and berry weight. Previous studies have also shown that the NDVI is often positively correlated with yield and berry weight [62,63,64]. The number of clusters showed less correlation with the NDVI through our sites (Table 1). In general, there were only weak correlations and temporal stability observed between the NDVI and berry composition data (Table 1). Although the NDVI is not the best indicator of variability in fruit composition, the evidence for an inverse correlation between phenols/anthocyanins level and the NDVI was still seen in three sites (sites 1, 4, and 5) through the years. Berry composition variables, especially phenolic accumulation, were inversely related to vigorous leaf canopies due to the impacts of fruit exposure to sunlight and flavonoid biosynthesis [54,65,66].

A thermal sensor is used in precision viticulture to measure the thermal energy emitted from plants, primarily to detect water stress, because leaf temperature increases with greater water stress [52]. The utilization of remote sensing thermal imagery in deficit irrigation has also been studied [67]. Image analysis using thermal imagery could distinguish differences in leaf water potential from the different irrigation treatments and acquisitions under full canopy growth, and dry conditions were most effective for monitoring the deficit irrigation [68]. Interestingly, a strong trend for a positive correlation between the thermal emission data and the pH was seen throughout the sites in 2016 (hot and dry year, Appendix A Figure A1), and thus, the use of remote sensing thermal emission data may be an effective method to determine pH levels under water stress conditions. However, this method requires further investigation due to its limited spatial and temporal consistency. The thermal data also indicated a positive correlation with the level of phenols at three sites and with the levels of anthocyanins at two sites in the dry year. Grape quality, particularly that of red grape varieties, is largely dependent on secondary metabolites, such as the accumulation of phenols and anthocyanins [15]. The lack of vine water influx reduces vegetative and reproductive growth, while increasing color intensity [19,68]. Even though many scientists agree that moderate water deficits are beneficial for grape quality [69,70,71], there is no clear general guideline for what level of water stress can be assigned to the moderate water deficit, because it depends on the site, variety, and cultural practice of the specific vineyard. However, remote sensing thermal imaging could be a tool to discover the balanced grapevine water level, optimum pH level, and zone of high phenols and anthocyanins for quality wine production.

The results also indicated that the site yield/berry compositions and their detection by remote sensing technologies were affected by annual climate. The dryer year (2016) showed more variations in yield and berry compositions and more capability of detecting the correlation between the variables than those in regular growing seasons (2015). Thus, the result was hypothesized that lower water content and subsequent stress could be a limiting factor for the reproductive process, affecting remote sensing’s detectability of variation in yield and fruit quality. Vine size has been demonstrated to significantly affect berry composition and yield in previous studies [22,24], and with stomata operating at full capacity most of the time under water stress, biomass production is linearly correlated with water consumption [1]. The vineyard water availability is largely affected by the microclimate within a single vineyard block, resulting in differences in the soil types, slope, sun exposure, and so on [15,18,19,20,21,22]. Previous research also confirmed that water restrictions in a vineyard increased spatial heterogeneity in the water status [72].

For the other vegetation indices measured in 2016 only, the CI green and GNDVI showed the most capability of detecting variation in the yield in most of the sites, with significant positive correlations. Both the CI green and GNDVI are derived from the ratio of green and NIR portion of the electromagnetic reflectance spectra. There are many factors contributing to reflectance in electromagnetic fields, but plant pigments are usually responsible for green reflectance, while a change in leaf structure or thickness might affect leaf reflectance in the NIR range [40]. Plant growth and reproduction are influenced by chlorophyll, which absorbs light energy and converts it to energy for photosynthesis, and red reflectance could be a reliable measure of photosynthesis activity in plants due to chlorophyll’s strong absorption of red radiation [42]. However, red reflectance measurements are very sensitive to solar irradiance, the presence of other pigment molecules, such as anthocyanins and carotenoids, and background interruption from soil reflectance [40,43,44]. Reflectance from green and NIR regions is relatively easy to detect in the plant canopy, since they are less influenced by these variables [36,51]. Previous research has found that these indices were a reliable and consistent indicator of canopy chlorophyll and nitrogen levels in plant leaves [48,51,73,74,75]. Changes in the leaf nitrogen concentration alter the photosynthetic membranes that are predominantly made up of chlorophyll [76]. There has been extensive evidence that leaf chlorophyll concentrations and nitrogen levels are associated, as are nitrogen levels and productivity in other crops [73,76,77,78]. The correlations between the indices from the RPAS flight and berry composition data were less significant than those between the indices and yield components. However, the CI green and anthocyanins exhibited noteworthy negative correlations in the two sites.

When using remote sensing data to perform the zonal management of grape quality and/or yield, it is necessary to produce reliable and precise maps that illustrate areas to implement zonal vineyard management. In Moran’s index analysis, the yield/berry composition data and data from the RPAS were highly clustered in 2016 (dry year), and the interpolated maps between the remote sensing data and the measured variables in this study displayed a similar spatial correlation pattern to its statistical one, which confirmed the compatibility of the spatial map analysis over other statistical tools for correlation basis analysis. Remote sensing maps can be used to pinpoint target areas in vineyards for precision viticulture applications, such as selective harvesting, better fruit exposure, and yield monitoring.

A keynote raised by this study was whether soil differences and management practices across vineyards could affect the capability of remote sensing to detect yield variations and quality changes. As a result of largescale glacial movement in the Niagara region, the research sites are planted on different soil types from moderately well drained Chinguacousy to poorly drained Beverly/Toledo soils that can differ considerably in soil properties [79]. In order to improve the correlation between the remote sensing data and variables indicating grape productivity, it is necessary to establish an analysis of soil profiles and its soil drainage capacity prior to the remote sensing data collection. A canopy-based remote sensing measurement can also be greatly affected by cultural practice, such as pruning and training systems [55]. A variety of cultural practices, such as training systems, water, floor, and canopy management, were also observed in the vineyards studied. The spur-pruned cordon systems with stronger horizontal profiles in site 3 could be more appropriate for airborne monitoring. In this study, the number of clusters showed less correlation with remote sensing data through the sites, which could be caused by the cluster thinning practice, as all six sites performed cluster thinning in the growing seasons, which could cause a more uniform number of clusters throughout the vineyard. Further investigation of the relationships between the remote sensing detectability and the vineyard cultural practices will promote the accuracy and usefulness of the remote sensing data to identify variabilities in spectral the behavior of grapevine canopy and will expand the knowledge of the spatiotemporal dynamics of plant physiology in vineyards.

## 4. Materials and Methods

### 4.1. Site Selection

This research involved six Cabernet franc vineyard blocks located within the Niagara Peninsula of Ontario, Canada. Several Niagara sub-appellations were represented in this study as follows: site 1: Buis vineyard in Four Mile Creek, site 2: Cave Spring vineyard in Beamsville Bench, site 3: Chateau des Charmes vineyard in St. David’s Bench, site 4: George vineyard in Lincoln Lakeshore, site 5: Kocsis vineyard in Lincoln Lakeshore, site 6: Pond View Vineyards in Four Mile Creek. For differences in site vineyard managements and block shapes and sizes, refer to Appendix A Table A1. Within each vineyard, a grid of geolocated sentinel vines (72–81 vines) was plotted in an 8 m × 8 m grid located via Invicta 115 GPS receiver (Raven Industries, Sioux Falls, SD, USA), which provided a 1 to 1.4 m accuracy and was improved further with a subsequent adjustment with the Port Weller, Ontario base location, resulting in a closing precision of 30 to 50 cm.

### 4.2. Electromagnetic Reflectance Data Collection by Multispectral Sensor on RPAS

The RPAS of the eBee Classic from the Parrot group in Switzerland was flown during veraison in 2015 and 2016 with an altitude of 90 m and a maximum speed of 60 km/h. A set of Sequoia multispectral sensors and a set of Sequoia Thermomap sensors (Parrot Group, Zug, Switzerland) were selected for gaining spectral data, the former equipped with an incident light sensor operating at a resolution of 1.2 megapixels (1280 × 960 pixels), a pixel size of 3.75 μm, representing a resolution of 8.47 cm at 90 m altitude in the visible and NIR region of reflectance with four wide bands (green: 530–570 nm, red: 640–680 nm, red edge: 730–740 nm, and the near infrared: 770–810 nm) and the latter analyzing the thermal–infrared spectrum range emission, covering 7000 to 16,000 nm at a resolution of 0.3 megapixels (640 × 512 pixels), a pixel size of 17 μm, representing a resolution of 17 cm at 90 m altitude. Additionally, the aircraft featured a GPS receiver, radiation monitor estimating the inbound radiation, and inertial system for maintaining the alignment and positioning of imaging. The RPAS also had an autopilot system that provided a visual range of 1000 m and a radio range of 5 km. The vehicle was powered by an electric motor with a battery life of 50 min. Air-Tech Solutions, Inverary, ON, Canada provided the RPAS and its ground control station for real-time tracking and the collection of images over each vineyard patch. Based on the data from the inertial station and radiation sensor, geometric and imaging adjustments were performed for geometry, reflectivity, image distortions, sun exposures, and vignetting effects in radiometric. A geometric correction using ground control points was performed to adjust the geometry of the image and adjust the bidirectional reflectance for ensuring the accuracy and consistency of the data. Geometric distortions caused by changes in RPAS attitude and altitude were corrected using the information provided by the inertial station. Radiometric correction was performed to correct the effects of vignetting. The data were also adjusted for the input of the sunshine sensor before VI generation. The NDVI and other indices were calculated from the mosaics assembled from the images acquired on each phase of each flight by choosing the overlapping pixels near nadir to minimize the problems of angle distortion and directional effects during the images’ acquisition.

### 4.3. Yield and Fruit Composition Data Acquisition

#### 4.3.1. Yield

The yield data were collected in 2015 and 2016 by harvesting sentinel grape vines as close as possible to the commercial harvesting dates. Vines were handpicked into plastic containers, and it was recorded how many clusters were harvested per vine. The weight of containers from each vine was measured with a field scale to calculate the amount (kg) of yield per vine. To determine the mean berry weight, sample clusters were stored at −25 °C, and 100 berries were weighed to determine their mean weight (g).

#### 4.3.2. Brix, pH, and Titratable Acidity

Frozen grape samples were heated in the Isotemp 228 heated water incubator from Fisher Scientific (Mississauga, ON, Canada) for 30 min at 85 °C juicing and filtering. Juice pH, Brix, and titratable acidity were determined with the Model 25 pH meter from Denver Instrument Inc. (Denver, CO, USA), Abbé refractometer model 10,450 from American Optical (Buffalo, NY, USA), and the centrifuged juice acquired from the IEC Centra CL2 from International Equipment Company (Needham Heights, MA, USA) was titrated to pH 8.2 with 0.1 NaOH via PC automatic titrator from Man-Tech (Guelph, ON, Canada) to determine titratable acidity (TA).

#### 4.3.3. Total Phenols

A micro method of Folin–Ciocalteu reagent was applied to calculate the total phenol contents in the grape juice samples [80,81]. After diluting the juice 10-fold, 20 μL of this mixture was transferred to the solution of 1.58 mL of water with 100 μL of Folin–Ciocalteu reagent from Sigma Aldrich (St. Louis, MO, USA) and heated for 8 min. In the next steps, 300 μL of sodium carbonate solution (NaCO3) was poured, blended again, and left in the dark at 20 °C for 2 h. The absorbance at 765 nm was then obtained from the solution, and from the standard curve, the phenol concentration was calculated.

#### 4.3.4. Total Anthocyanins

A pH shift technique was applied to quantify anthocyanins in the grapes [82]. Juice samples were diluted with 9 mL of each buffer solution (pH 1.0 and pH 4.5) and left to equilibrate for an hour in the dark prior to measurement with a 2100 pro UV/Vis spectrophotometer from Biochrom Ltd. (Cambridge, UK). According to the given formula [82], the anthocyanin concentration (mg/L) was determined as A520 (pH 1.0–pH 4.5) × 255.75.

### 4.4. Mapping and Data Analysis

#### 4.4.1. Vegetation Indices (VIs) Extraction

Several VIs are applied in viticulture to measure a variety of plant characteristics, for example canopy size and structure, leaf color intensity, area index of leaves, plant physiology, and nutrient deficiency [45,47]. Numerous remote sensing indices can be calculated from the green, red, red edge, and NIR regions of EM reflectance to detect reproductive growth and potential grape quality. Therefore, the feature indices (Table 4) in the study were used to characterize the plant yield and berry composition, according to previous studies [48,49,50,51].

#### 4.4.2. Mapping

The spatial distribution of the remote sensing data for each site was displayed using ArcMap 10.6, and the maps of the point data collected from the sentinel vines were created via the inverse-distance-weighted (IDW) interpolation method [83]. The data symbology was mapped using Quantile breaks, which avoided extraordinarily large or empty classes and made map interpretation easier. Data from the NDVI, yield, and fruit compositions were imported and displayed in ArcMap 10.6 using the World Geodetic System 1984 as well as projected in Universal Transverse Mercator zone 17N.

#### 4.4.3. Moran’s Global Index: Spatial Autocorrelation

To evaluate the feasibility of implementing precision viticulture by using remote sensing data, it is necessary to investigate the spatial distribution of the variables. Clustered variables might be better for precision viticulture applications, as they can easily be targeted at larger vineyard clusters or zones, rather than scattered throughout the vineyard. Spatial autocorrelations were performed using the autocorrelation tool, Moran’s global index, to determine whether a pattern expressed was clustered, dispersed, or random.

#### 4.4.4. Correlation-Based Analyses

Correlation statistics were carried out through XLSTAT v2021. The Shapiro–Wilk test was applied to each data to verify normality, and any outliers were highlighted on boxplots after careful evaluation of the data variation.

Principle componence analysis (PCA).

With PCA, a complex dataset can be reduced to simple data that still contains most of the original data [84]. Using PCAs, correlated variables are compressed into new uncorrelated variables called principal components. The first components contain the majority of the information within the initial variables. Upon removing the effect of the first component, the second component explains the most variance, and so on. The PCAs produced correlation circles, which reflect the projection of variables into the linear space of the factors. Prior to the next step to perform a PCA, data were standardized. Prior to using the first two factors, the first three factors were assessed to see if they explained the greatest variation in the data. Additionally, each variable was tested for its square cosine across the first two factors in order to confirm its relationship with the axis. A short vector of the correlation circle (the square cosine value is close to 0) indicates that the chosen factors/PCA model does not adequately explain the variable. It can then be used to visualize correlations between many different variables by using a correlation circle, where variables located far from the center and those located close together are correlated positively, but variables at opposite sides indicate a negative correlation. A PCA was run on each site and year separately to determine the key groupings of remote sensing data and vineyard variables.

Pearson’s correlation coefficient (PCC)

PCC measures how strongly and in which direction the two variables are linearly related. In a correlation coefficient with range of −1 to +1, −1 indicates the perfect negative correlation, +1 indicates the perfect positive correlation, and 0 indicates no relation. The PCC matrices are also used to confirm the PCA results, because highly correlated variables in PCA can pervert the vectors of other variables and lower the values of other potential relationships. The PCC was computed on all vintage and site data at a 95% confidence level to find a meaningful relationship between the remote sensing data and vineyard variable data.

## 5. Conclusions

According to the findings of this study, the data analysis from remote sensing technology has some potential for predicting grape quality and productivity, but the site and growing season conditions can limit its reliability. A strong trend of a positive correlation of the NDVI with the yield and berry weight was seen throughout the sites and years. The CI green and GNDVI also showed a high level of capability for detecting variation in the yield, and the RVI was a good indicator of berry weight. The other strong trend was for a positive correlation between the thermal emission data and berry pH, seen throughout the sites and years, especially in 2016, which was a dry growing season. The thermal data also indicated a positive correlation with the level of phenols at three sites and with the level of anthocyanins at two sites in the dry year. Even though this research confirmed the possible application of thermal imaging data in monitoring the fruit quality, further research will be required to confirm the impacts of water stress on the remote sensing of fruit quality. This study also confirmed that the spatial patterns and the clustering of variables demonstrated the potential use of remote sensing maps to pinpoint target areas and to perform zonal management for selective harvesting in a single vineyard.

## Figures and Tables

**Figure 1 plants-14-00088-f001:**
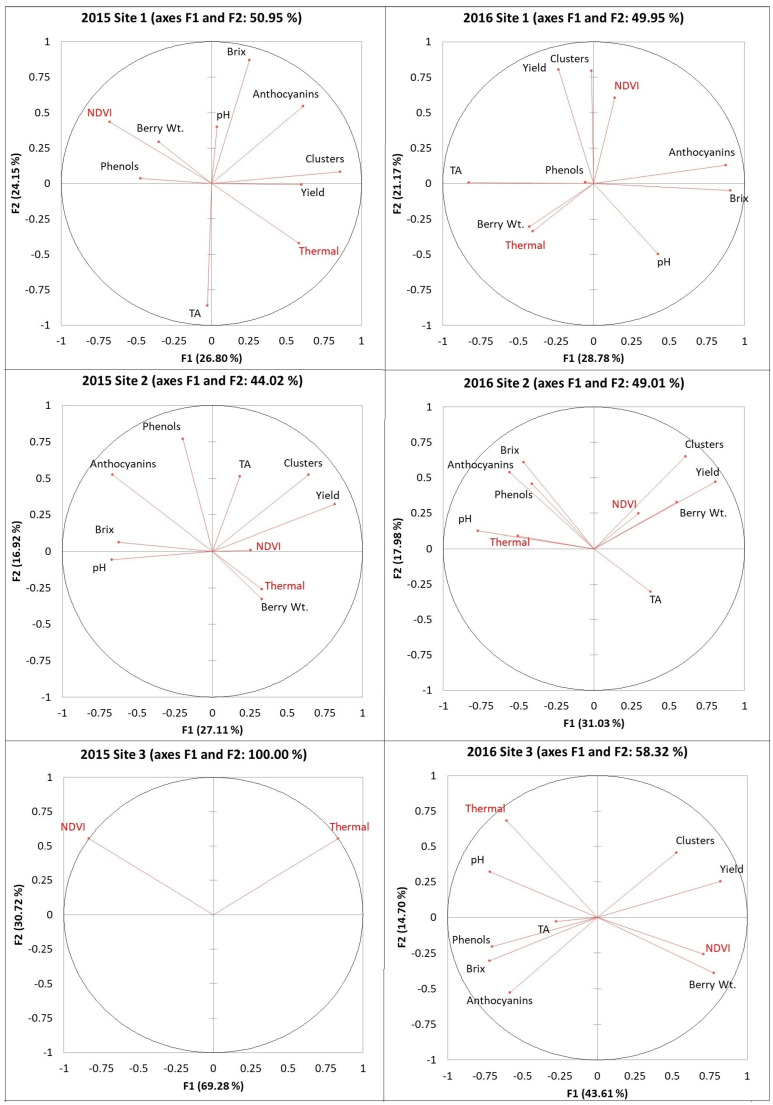
PCA results of remote sensing, vineyard yield, and berry composition from sites 1, 2, and 3 in 2015 and 2016. No harvest data collected at site 3 vineyard in 2015. Abbreviations: NDVI = normalized difference vegetation index, Thermal = thermal emission data, Clusters = number of clusters, Berry WT = berry weight, TA = titratable acidity.

**Figure 2 plants-14-00088-f002:**
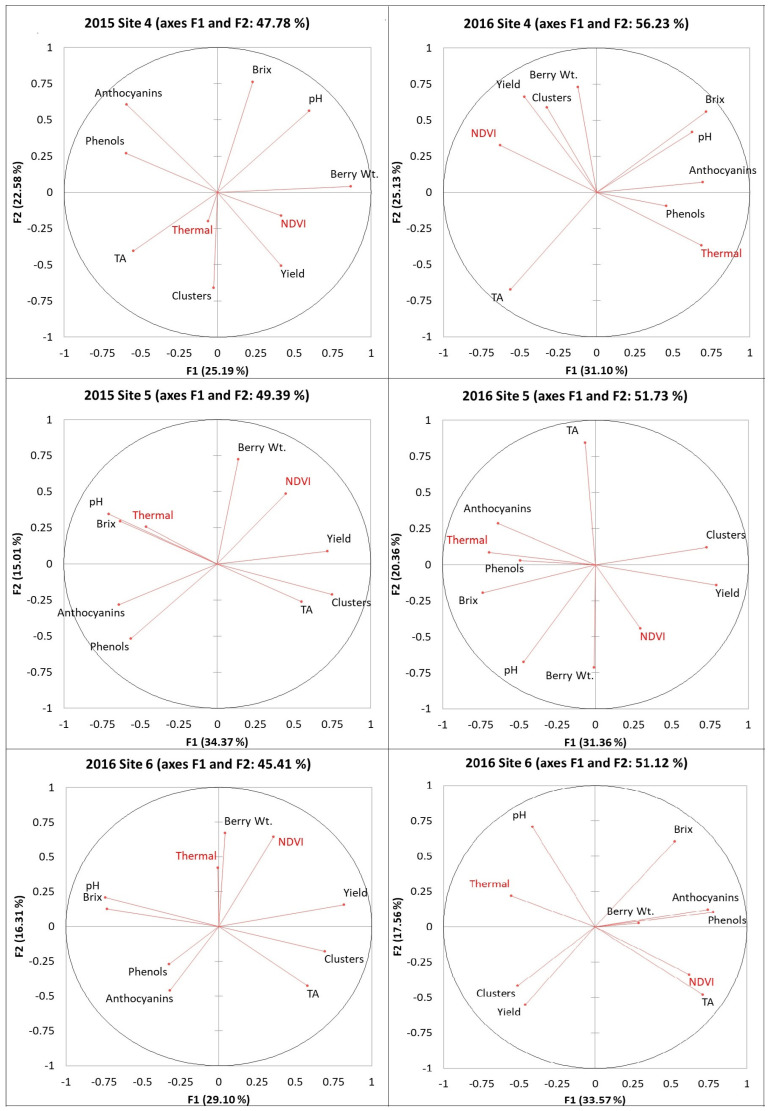
PCA results of remote sensing, vineyard yield, and berry composition from sites 4, 5, and 6 in 2015 and 2016. Abbreviations: NDVI = normalized difference vegetation index, Thermal = thermal emission data, Clusters = number of clusters, Berry WT = berry weight, TA = titratable acidity.

**Figure 3 plants-14-00088-f003:**
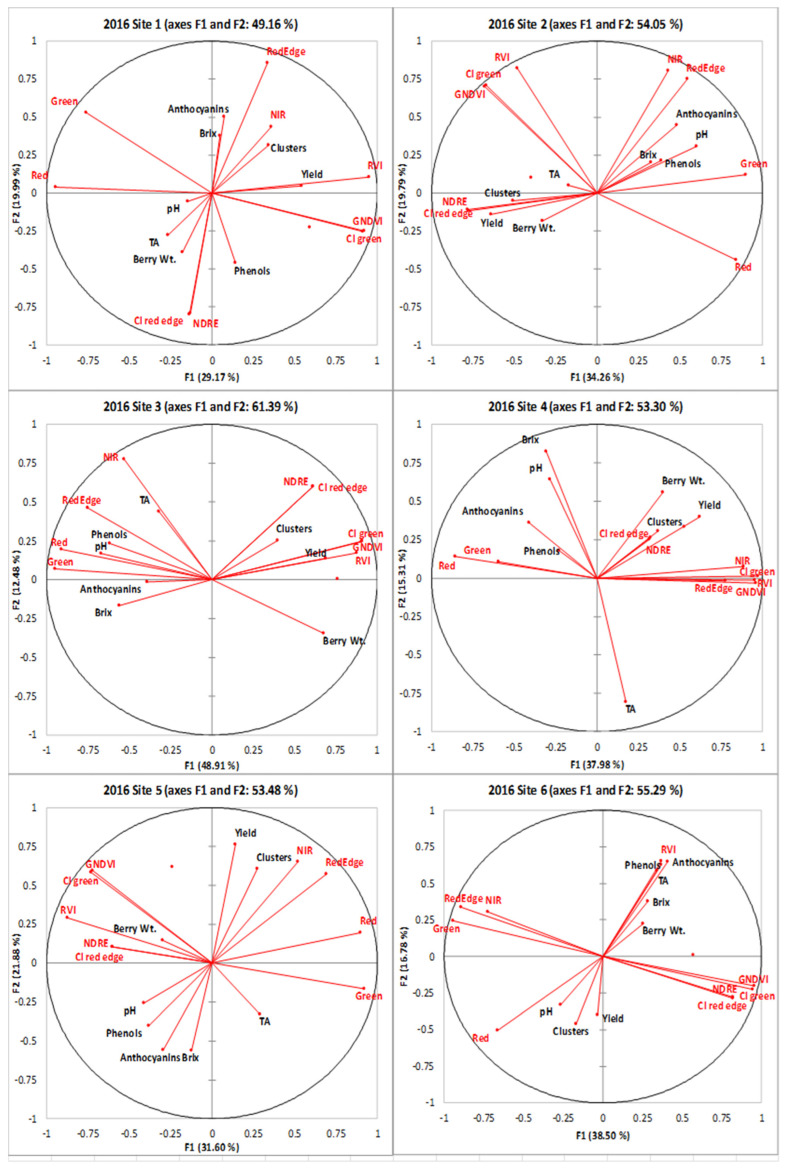
PCA results among indices from the RPAS flight, vineyard yield, and berry composition in six Niagara vineyards from 2016. Variables include data in six Ontario vineyards in 2016. Abbreviations: Berry WT = berry weight, TA = titratable acidity, CI green = green chlorophyll index, CI red edge = red edge chlorophyll index, NDRE = red edge normalized difference vegetation index, GNDVI = NDVI green, RVI = ratio vegetation index.

**Figure 4 plants-14-00088-f004:**
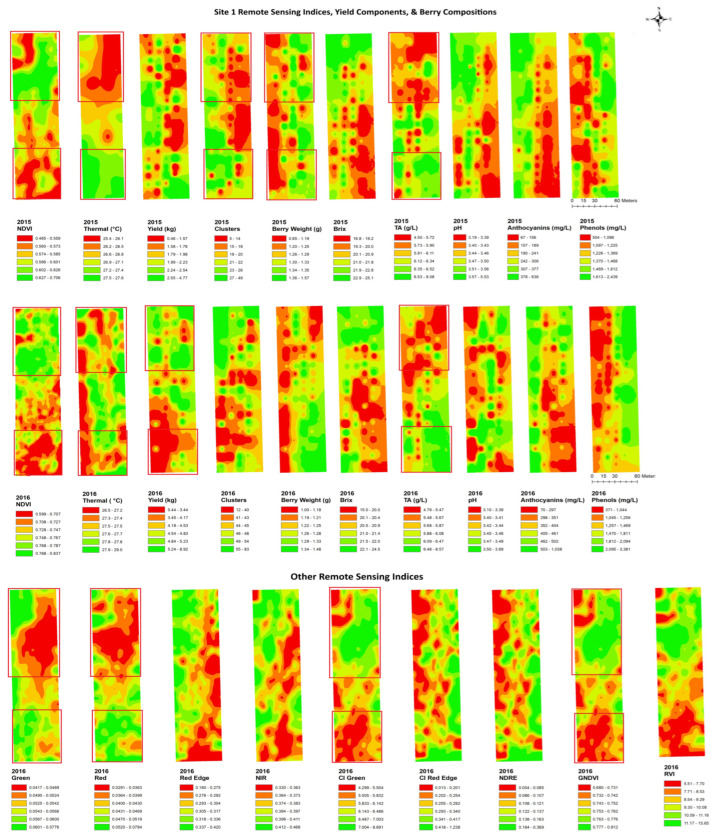
Spatial maps of data from RPAS flight, vineyard yield, and berry composition in site 1 from 2015 and 2016. Abbreviations: Berry WT = berry weight, TA = titratable acidity, NDVI = normalized difference vegetation index, Thermal = thermal emission data, CI green = green chlorophyll index, CI red edge = red edge chlorophyll index, NDRE = red edge normalized difference vegetation index, GNDVI = NDVI green, RVI = ratio vegetation index.

**Figure 5 plants-14-00088-f005:**
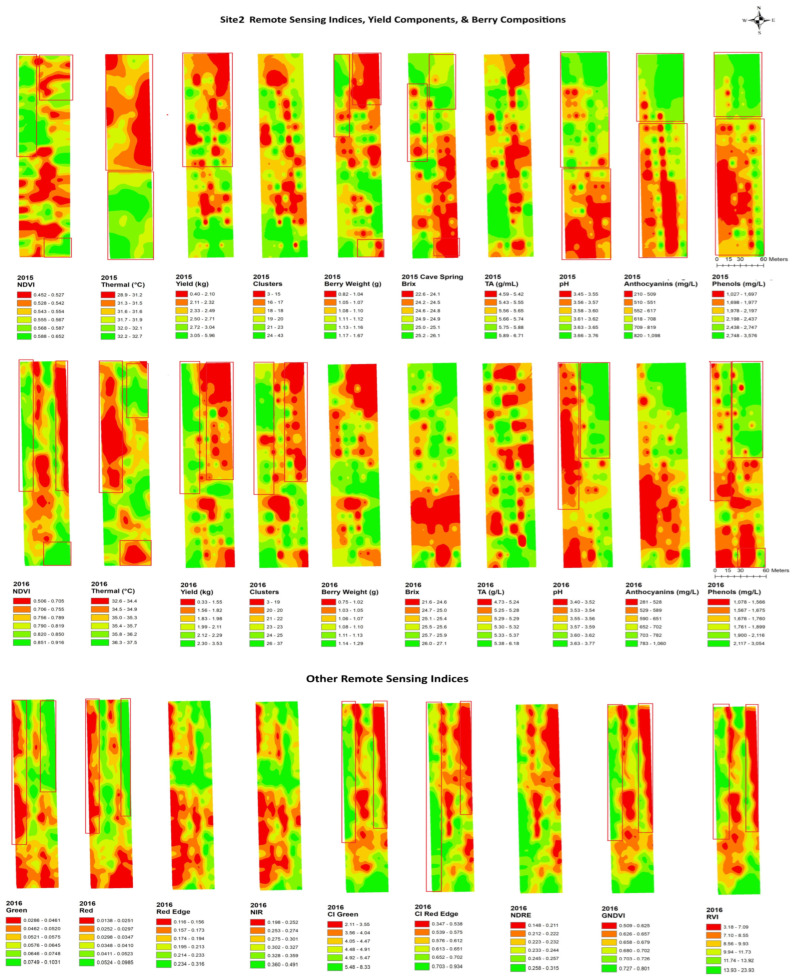
Spatial maps of data from RPAS flight, vineyard yield and berry composition in site 2 from 2015 and 2016. Abbreviations: Berry WT = berry weight, TA = titratable acidity, NDVI = normalized difference vegetation index, Thermal = thermal emission data, CI green = green chlorophyll index, CI red edge = red edge chlorophyll index, NDRE = red edge normalized difference vegetation index, GNDVI = NDVI green, RVI = ratio vegetation index.

**Figure 6 plants-14-00088-f006:**
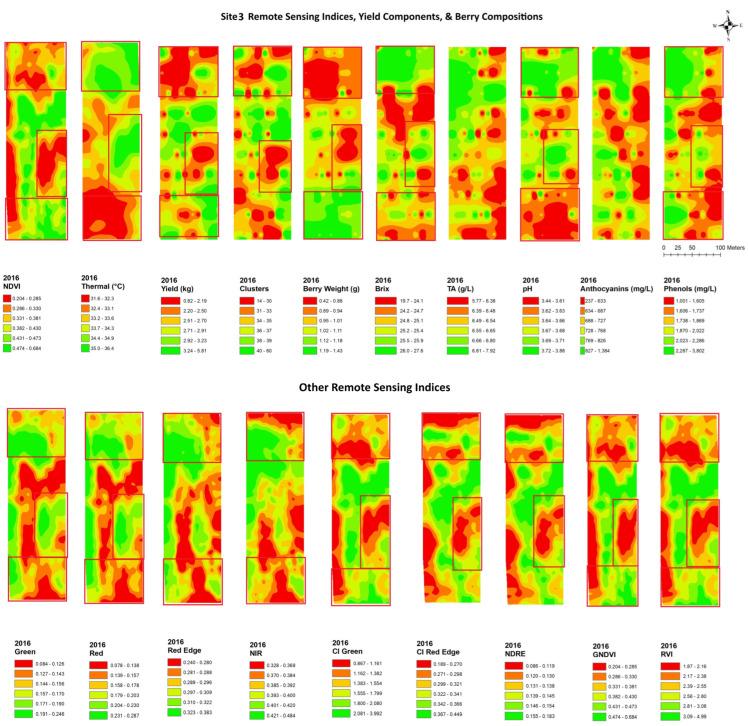
Spatial maps of data from RPAS flight, vineyard yield, and berry composition in site 3 from 2016. Abbreviations: Berry WT = berry weight, TA = titratable acidity, NDVI = normalized difference vegetation index, Thermal = thermal emission data, CI green = green chlorophyll index, CI red edge = red edge chlorophyll index, NDRE = red edge normalized difference vegetation index, GNDVI = NDVI green, RVI = ratio vegetation index.

**Figure 7 plants-14-00088-f007:**
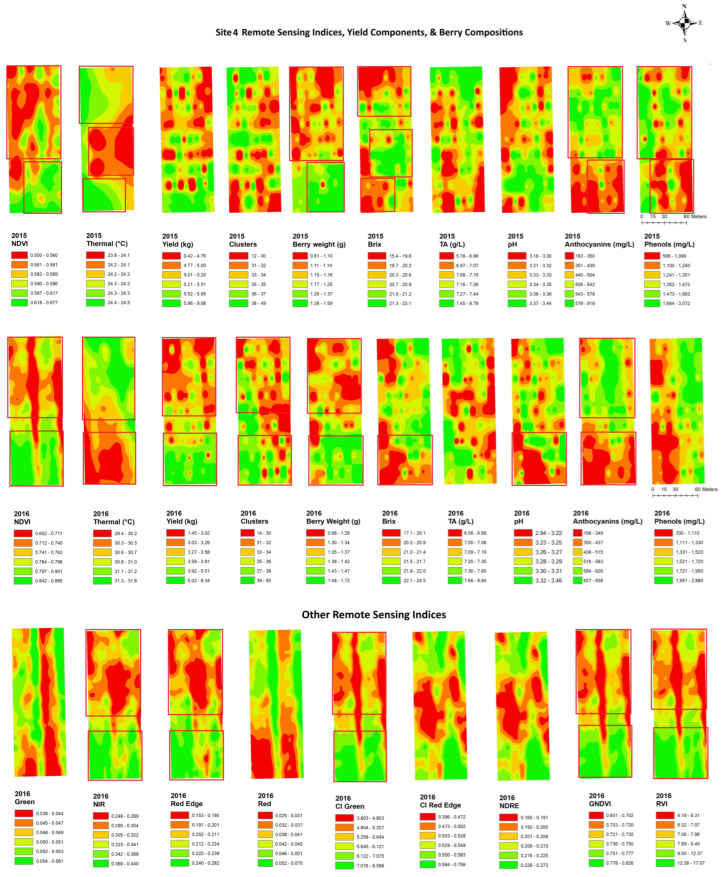
Spatial maps of data from RPAS flight, vineyard yield, and berry composition in site 4 from 2015 and 2016. Abbreviations: Berry WT = berry weight, TA = titratable acidity, NDVI = normalized difference vegetation index, Thermal = thermal emission data, CI green = green chlorophyll index, CI red edge = red edge chlorophyll index, NDRE = red edge normalized difference vegetation index, GNDVI = NDVI green, RVI = ratio vegetation index.

**Figure 8 plants-14-00088-f008:**
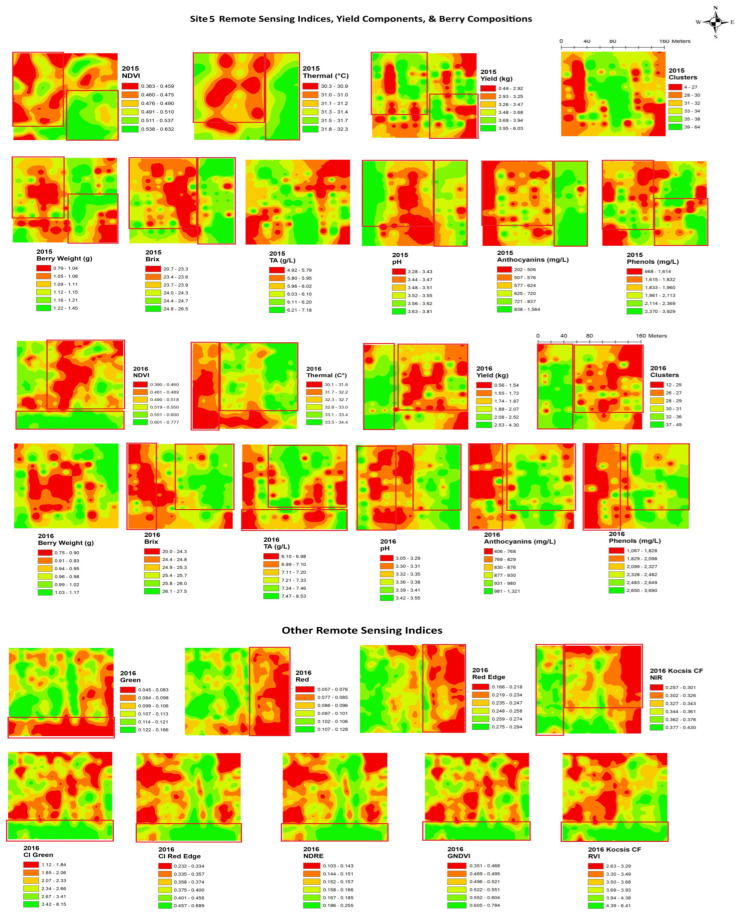
Spatial maps of data from RPAS flight, vineyard yield, and berry composition in site 5 from 2015 and 2016. Abbreviations: Berry WT = berry weight, TA = titratable acidity, NDVI = normalized difference vegetation index, Thermal = thermal emission data, CI green = green chlorophyll index, CI red edge = red edge chlorophyll index, NDRE = red edge normalized difference vegetation index, GNDVI = NDVI green, RVI = ratio vegetation index.

**Figure 9 plants-14-00088-f009:**
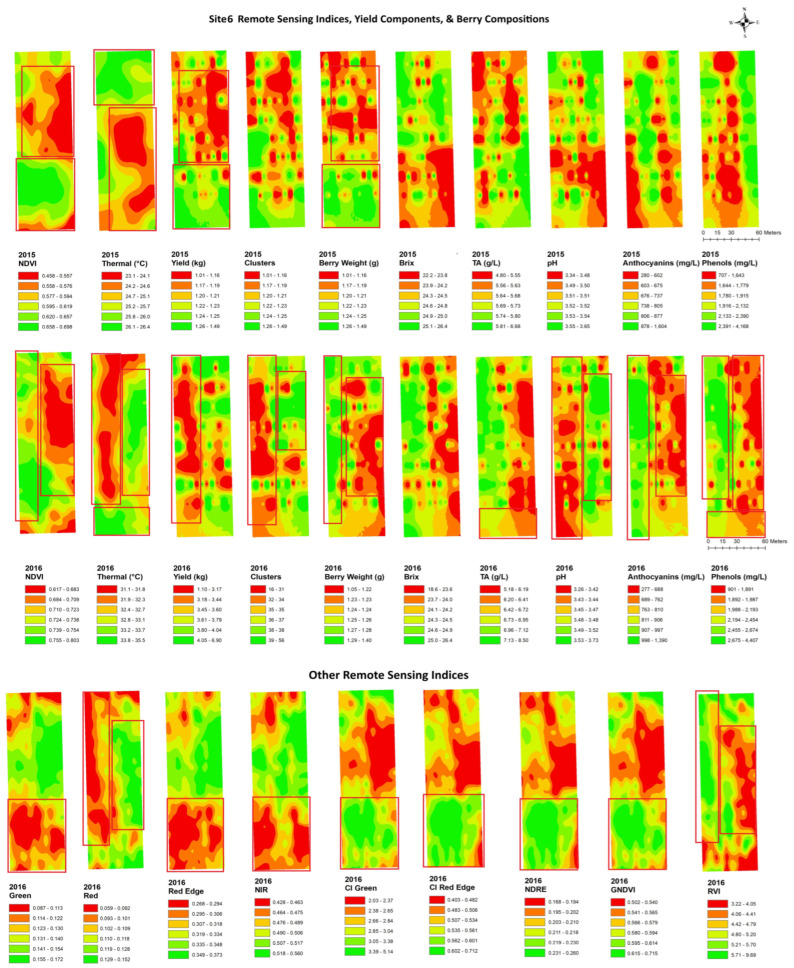
Spatial maps of data from RPAS flight, vineyard yield, and berry composition in site 6 from 2015 and 2016. Abbreviations: Berry WT = berry weight, TA = titratable acidity, NDVI = normalized difference vegetation index, Thermal = thermal emission data, CI green = green chlorophyll index, CI red edge = red edge chlorophyll index, NDRE = red edge normalized difference vegetation index, GNDVI = NDVI green, RVI = ratio vegetation index.

**Table 1 plants-14-00088-t001:** Pearson’s correlation results between remote sensing NDVI vs. yield and berry composition data in six Niagara vineyards from 2015 and 2016. Significant variables (95% confidence) were listed in color, with blank cells representing no correlation: blue boxes = positive correlation, red boxes = negative correlation, black boxes = no data collected. Abbreviations: NDVI = normalized difference vegetation index, Thermal = thermal emission data, Clusters = number of clusters, Berry Wt = berry weight, TA = titratable acidity, and Antho = anthocyanins.

Remote Sensing NDVI (Correlation Matrix)									
Site	Year	Clusters	Yield	Berry Wt.	Brix	pH	TA	Phenols	Antho
**1**	**2015**	** −0.392 **	−0.131	** 0.264 **	0.129	0.055	−**0.272**	0.220	−0.222
	**2016**	0.226	** 0.410 **	−0.197	0.042	−0.157	−**0.247**	0.022	0.091
**2**	**2015**	0.049	0.181	** 0.238 **	−**0.238**	−0.002	0.107	−0.053	−0.047
	**2016**	** 0.243 **	** 0.241 **	0.033	−0.032	−0.064	0.095	−0.047	0.095
**3**	**2015**								
	**2016**	** 0.304 **	** 0.512 **	** 0.462 **	−**0.432**	−**0.513**	−0.213	−**0.355**	−0.220
**4**	**2015**	0.035	0.117	** 0.328 **	−0.021	0.131	0.055	−0.181	−**0.232**
	**2016**	** 0.252 **	** 0.390 **	** 0.297 **	−**0.262**	−0.195	0.148	−0.176	−**0.263**
**5**	**2015**	0.137	** 0.233 **	** 0.293 **	−0.107	−**0.294**	** 0.268 **	−**0.384**	−0.192
	**2016**	0.134	** 0.234 **	0.141	−0.105	0.031	−**0.328**	0.059	−0.151
**6**	**2015**	0.056	** 0.309 **	** 0.362 **	−0.162	−0.142	0.089	−0.124	−0.202
	**2016**	−**0.220**	0.023	** 0.321 **	0.164	−**0.296**	** 0.512 **	** 0.294 **	** 0.406 **
**Remote Sensing Thermal (Correlation Matrix)**									
**Site**	**Year**	**Clusters**	**Yield**	**Berry Wt.**	**Brix**	**pH**	**TA**	**Phenols**	**Antho**
**1**	**2015**	** 0.300 **	0.143	−0.083	−0.130	−0.035	0.218	−0.201	0.087
	**2016**	−0.119	0.032	** 0.352 **	** −0.238 **	** 0.247 **	0.112	0.160	−**0.438**
**2**	**2015**	0.029	0.053	0.097	−0.090	** −0.354 **	0.124	−0.186	−0.216
	**2016**	−0.085	−0.194	−0.097	0.034	** 0.485 **	−0.212	** 0.254 **	0.177
**3**	**2015**								
	**2016**	−0.093	−**0.348**	−**0.712**	** 0.233 **	** 0.539 **	0.145	0.196	0.047
**4**	**2015**	−0.106	−0.003	0.096	−**0.281**	−0.115	0.014	0.155	−0.165
	**2016**	−**0.236**	−**0.451**	−**0.284**	0.167	** 0.265 **	−0.095	** 0.351 **	** 0.411 **
**5**	**2015**	−**0.299**	−0.171	0.007	** 0.299 **	** 0.310 **	−0.162	0.094	** 0.346 **
	**2016**	−**0.381**	−**0.470**	0.004	** 0.427 **	** 0.277 **	0.162	** 0.303 **	** 0.237 **
**6**	**2015**	0.056	0.171	0.039	0.180	0.085	−0.118	0.018	−0.015
	**2016**	0.139	0.132	−0.181	−0.184	** 0.311 **	−**0.337**	−**0.355**	−**0.262**

**Table 2 plants-14-00088-t002:** Pearson’s correlation results between indices from the RPAS flight vs. yield and berry composition data in six Niagara vineyards from 2016. The indices from the RPAS flight included green, red, red edge, NIR, CI green, CI red edge, NDRE, GNDVI, and RVI. Those variables with significant (95% confidence) were listed in color, with blank cells representing no correlation: blue boxes = positive correlation with indices, red boxes = negative correlation with indices. Abbreviations: Berry WT = berry weight, TA = titratable acidity, Antho = anthocyanins, CI green = green chlorophyll index, CI red edge = red edge chlorophyll index, NDRE = red edge normalized difference vegetation index, GNDVI = NDVI green, RVI = ratio vegetation index.

Variables	Vineyards	Clusters	Yield	Berry Wt.	Brix	pH	TA	Phenols	Antho
Cl green	Site 1	0.059	** 0.272 **	−0.165	−0.002	−0.078	** −0.231 **	0.162	0.017
	Site 2	0.222	** 0.253 **	0.025	−0.084	−0.192	0.144	−0.158	−0.042
	Site 3	** 0.314 **	** 0.538 **	** 0.472 **	** −0.462 **	** −0.518 **	−0.186	** −0.373 **	** −0.239 **
	Site 4	** 0.267 **	** 0.523 **	** 0.345 **	** −0.242 **	−0.227	0.118	−0.129	** −0.285 **
	Site 5	0.130	** 0.242 **	0.158	−0.073	0.064	** −0.338 **	0.070	−0.129
	Site 6	−0.088	0.017	0.163	0.147	−0.167	0.207	0.159	0.199
Cl red edge	Site 1	** −0.338 **	** −0.265 **	0.184	−0.021	0.184	0.021	0.218	−0.183
	Site 2	0.204	** 0.308 **	0.055	** −0.293 **	** −0.385 **	−0.027	** −0.228 **	** −0.440 **
	Site 3	** 0.227 **	** 0.344 **	0.169	** −0.263 **	** −0.221 **	−0.048	** −0.270 **	−0.135
	Site 4	0.097	** 0.386 **	0.085	−0.093	−0.026	0.061	0.050	−0.060
	Site 5	−0.110	−0.070	−0.035	−0.006	−0.002	−0.004	** 0.247 **	0.189
	Site 6	−0.047	−0.035	0.093	0.205	−0.050	0.120	0.159	0.157
GNDVI	Site 1	0.075	** 0.302 **	−0.165	−0.007	−0.084	−0.224	0.176	0.023
	Site 2	** 0.244 **	** 0.280 **	0.050	−0.052	−0.202	0.124	−0.126	−0.017
	Site 3	** 0.339 **	** 0.541 **	** 0.477 **	** −0.441 **	** −0.527 **	−0.187	** −0.371 **	−0.212
	Site 4	** 0.244 **	** 0.474 **	** 0.330 **	** −0.245 **	−0.216	0.120	−0.111	** −0.251 **
	Site 5	0.134	** 0.234 **	0.141	−0.105	0.031	** −0.328 **	0.059	−0.151
	Site 6	−0.102	0.011	0.199	0.179	−0.152	0.217	0.186	0.218
Green	Site 1	−0.035	** −0.319 **	0.033	0.138	0.112	0.104	** −0.378 **	0.088
	Site 2	** −0.368 **	** −0.445 **	−0.212	0.159	** 0.506 **	−0.088	** 0.358 **	** 0.385 **
	Site 3	** −0.292 **	** −0.520 **	** −0.580 **	** 0.404 **	** 0.613 **	** 0.315 **	** 0.470 **	** 0.228 **
	Site 4	−0.070	** −0.289 **	−0.034	** 0.250 **	0.210	−0.106	−0.047	−0.098
	Site 5	0.111	0.027	−0.214	−0.120	** −0.219 **	** 0.225 **	** −0.249 **	0.054
	Site 6	0.109	−0.003	−0.166	−0.161	0.176	−0.191	−0.181	** −0.221 **
NDRE	Site 1	** −0.346 **	** −0.271 **	0.188	−0.023	0.195	0.021	0.222	−0.187
	Site 2	0.206	** 0.312 **	0.058	** −0.290 **	** −0.394 **	−0.024	** −0.235 **	** −0.436 **
	Site 3	** 0.225 **	** 0.343 **	0.170	** −0.266 **	** −0.221 **	−0.050	** −0.273 **	−0.136
	Site 4	0.090	** 0.379 **	0.079	−0.100	−0.034	0.067	0.048	−0.060
	Site 5	−0.111	−0.073	−0.042	−0.006	−0.004	−0.004	** 0.251 **	0.191
	Site 6	−0.052	−0.039	0.101	0.213	−0.049	0.124	0.167	0.164
NIR	Site 1	0.065	−0.018	** −0.228 **	0.216	0.045	** −0.227 **	** −0.334 **	0.176
	Site 2	** −0.233 **	** −0.279 **	−0.200	0.126	** 0.393 **	0.030	** 0.300 **	** 0.430 **
	Site 3	0.029	−0.161	** −0.502 **	0.104	** 0.473 **	** 0.446 **	** 0.435 **	0.131
	Site 4	** 0.303 **	** 0.469 **	** 0.437 **	−0.157	−0.150	0.083	−0.210	** −0.455 **
	Site 5	** 0.407 **	** 0.425 **	−0.148	** −0.398 **	** −0.364 **	−0.104	** −0.373 **	−0.166
	Site 6	0.080	−0.002	−0.039	−0.058	0.205	−0.083	−0.099	−0.162
Red	Site 1	** −0.226 **	** −0.455 **	0.133	0.029	0.192	0.196	−0.135	−0.033
	Site 2	** −0.353 **	** −0.381 **	−0.149	0.109	** 0.308 **	−0.100	0.221	0.147
	Site 3	** −0.238 **	** −0.474 **	** −0.570 **	** 0.380 **	** 0.600 **	** 0.349 **	** 0.457 **	** 0.232 **
	Site 4	−0.209	** −0.331 **	−0.197	** 0.291 **	0.206	−0.159	0.152	0.154
	Site 5	** 0.294 **	0.170	** −0.278 **	** −0.320 **	** −0.425 **	0.141	** −0.333 **	−0.092
	Site 6	** 0.237 **	−0.020	** −0.321 **	−0.184	** 0.357 **	** −0.520 **	** −0.326 **	** −0.452 **

**Table 3 plants-14-00088-t003:** Moran’s I analysis results (Moran’s index and *p*-value) for data from remote sensing, yield, and berry composition data in six Niagara vineyards from 2015 and 2016 (95% confidence): blue boxes = clustered, red boxes = random, yellow boxes = dispersed, black boxes = no data collected. Abbreviations: NDVI = normalized difference vegetation index, Thermal = thermal emission data, Clusters = number of clusters, Berry WT = berry weight, TA = titratable acidity.

Moran’s Index										
Vineyards	NDVI	Thermal	Clusters	Yield	Berry Wt.	Brix	pH	TA	Phenols	Anthocyanin
**2015 Site 1 (*n* = 76) *p*-value**	** 0.6284 **	** 0.8308 **	** 0.1833 **	** −0.0335 **	** 0.0072 **	** 0.2353 **	** 0.0022 **	** 0.2199 **	** 0.1478 **	** 0.4094 **
	** 0.0001 **	** 0.0001 **	** 0.0600 **	** 0.8480 **	** 0.8456 **	** 0.0201 **	** 0.8088 **	** 0.0269 **	** 0.1257 **	** 0.0000 **
**2016 Site 1 (*n* = 76) *p*-value**	** 0.7496 **	** 0.4990 **	** −0.0170 **	** 0.1490 **	** 0.2992 **	** 0.1879 **	** −0.0114 **	** 0.1785 **	** 0.4415 **	** 0.2608 **
	** 0.0001 **	** 0.0001 **	** 0.9712 **	** 0.1222 **	** 0.0030 **	** 0.0550 **	** 0.9854 **	** 0.0660 **	** 0.0000 **	** 0.0080 **
**2015 Site 2 (*n* = 75) *p*-value**	** 0.5505 **	** 0.6819 **	** −0.1493 **	** 0.0088 **	** 0.1385 **	** 0.1385 **	** 0.3476 **	** −0.0120 **	** −0.0640 **	** −0.1030 **
	** 0.0001 **	** 0.0001 **	** 0.3000 **	** 0.2328 **	** 0.2233 **	** 0.2328 **	** 0.0012 **	** 0.9905 **	** 0.7023 **	** 0.4991 **
**2016 Site 2 (*n* = 75) *p*-value**	** 0.2315 **	** 0.6762 **	** 0.1013 **	** 0.2407 **	** 0.3109 **	** 0.3583 **	** 0.4281 **	** −0.2295 **	** −0.0148 **	** 0.2929 **
	** 0.0225 **	** 0.0001 **	** 0.3799 **	** 0.0170 **	** 0.0023 **	** 0.0004 **	** 0.0001 **	** 0.0884 **	** 0.9923 **	** 0.0032 **
**2015 Site 3 (*n* = 80) *p*-value**	** 0.7450 **	** 0.7829 **								
	** 0.0001 **	** 0.0001 **								
**2016 Site 3 (*n* = 80) *p*-value**	** 0.8677 **	** 0.8585 **	** 0.1407 **	** 0.2960 **	** 0.6133 **	** 0.2864 **	** 0.3890 **	** 0.1844 **	** 0.4918 **	** 0.2477 **
	** 0.0001 **	** 0.0001 **	** 0.2182 **	** 0.0031 **	** 0.0001 **	** 0.0032 **	** 0.0002 **	** 0.1104 **	** 0.0001 **	** 0.0124 **
**2015 Site 4 (*n* = 72) *p*-value**	** 0.2412 **	** 0.7988 **	** −0.1248 **	** −0.1096 **	** 0.3223 **	** 0.0361 **	** 0.0390 **	** 0.0398 **	** 0.0227 **	** 0.1677 **
	** 0.0170 **	** 0.0001 **	** 0.4304 **	** 0.4920 **	** 0.0022 **	** 0.7187 **	** 0.7015 **	** 0.7015 **	** 0.7927 **	** 0.1955 **
**2016 Site 4 (*n* = 72) *p*-value**	** 0.4626 **	** 0.7691 **	**−0.3332**	** 0.2435 **	** 0.2018 **	** 0.1868 **	** 0.2503 **	** −0.0765 **	** 0.3122 **	** 0.3354 **
	** 0.0001 **	** 0.0001 **	**0.0230**	** 0.0152 **	** 0.1235 **	** 0.1529 **	** 0.0115 **	** 0.6511 **	** 0.0021 **	** 0.0024 **
**2015 Site 5 (*n* = 81) *p*-value**	** 0.3498 **	** 0.6784 **	** 0.2277 **	** −0.0273 **	** 0.2277 **	** 0.3498 **	** 0.4994 **	** 0.0706 **	** 0.1553 **	** 0.4684 **
	** 0.0006 **	** 0.0001 **	** 0.0247 **	** −0.0273 **	** 0.0248 **	** 0.0006 **	** 0.0001 **	** 0.4328 **	** 0.1167 **	** 0.0000 **
**2016 Site 5 (*n* = 81) *p*-value**	** 0.6561 **	** 0.7187 **	** 0.2992 **	** 0.2986 **	** 0.0902 **	** 0.3473 **	** 0.3050 **	** 0.1488 **	** 0.3043 **	** 0.2330 **
	** 0.0001 **	** 0.0001 **	** 0.0035 **	** 0.0034 **	** 0.3378 **	** 0.0007 **	** 0.0028 **	** 0.1300 **	** 0.0031 **	** 0.0219 **
**2015 Site 6 (*n* = 80) *p*-value**	** 0.6715 **	** 0.9481 **	** −0.0490 **	** 0.0044 **	** −0.1596 **	** 0.1275 **	** 0.0685 **	** −0.0822 **	** 0.4372 **	** 0.3143 **
	** 0.0001 **	** 0.0001 **	** 0.7837 **	** 0.8982 **	** 0.2698 **	** 0.2948 **	** 0.5415 **	** 0.5984 **	** 0.0001 **	** 0.0028 **
**2016 Site 6 (*n* = 80) *p*-value**	** 0.8527 **	** 0.6785 **	** 0.2690 **	** 0.2350 **	** 0.1146 **	** −0.0363 **	** 0.0296 **	** 0.3706 **	** 0.1987 **	** 0.3235 **
	** 0.0001 **	** 0.0001 **	** 0.0082 **	** 0.0207 **	** 0.3377 **	** 0.8559 **	** 0.7513 **	** 0.0001 **	** 0.1118 **	** 0.0021 **

**Table 4 plants-14-00088-t004:** Vegetation indices (VIs) used in this study to characterize the plant yield and berry composition.

Vegetation Indices	Equations
CI green (green chlorophyll index)	(NIR/Green) − 1
CI red edge (red edge chlorophyll index)	(NIR/RedEdge) − 1
NDRE (red edge normalized difference vegetation index)	(NIR-RedEdge)/(NIR + RedEdge)
GNDVI (NDVI green)	(NIR-Green)/(NIR + Green)
GRVI (green red vegetation index)	(Green-Red)/(Green + Red)
RVI (ratio vegetation index)	NIR/Red

## Data Availability

The data presented in this study are available on request from the corresponding author, because the data involves the geo-referenced site information of private landowners.

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
