# Peer review of "A Feasibility Study on Utilizing Remote Sensing Data to Monitor Grape Yield and Berry Composition for Selective Harvesting"

_plants, 2024, doi:10.3390/plants14010088_

Round 1
Reviewer 1 Report
Comments and Suggestions for Authors
1. Some sentences are overly complex. For example, sections discussing "thermal emission correlations" can be simplified, please add some explainations.
2. Please clarify statistical analysis: The PCA and Pearson correlation results are difficult to interpret for readers without a strong statistical background. Add simple explanations and a flowchart to illustrate the steps.
3. Please expand discussion on thermal data utility: Thermal imaging data is underexplored in this study. Discuss its potential applications in vineyard management, such as early water stress detection or pH optimization.
4. Address year-to-year variability: The NDVI correlations vary significantly between 2015 and 2016. Provide hypotheses, such as weather differences, that might explain these inconsistencies.
5. Authors should improve their visual data representation: The figures are informative but cluttered. Use consistent color schemes and add annotations to highlight key insights from spatial maps.
6. Please discuss more about methodological limitations: Including a subsection detailing how management practices and soil differences across vineyards may have influenced results would be better.
7. Strengthen practical implications: Please provide examples of how the findings can help vineyard managers implement precision irrigation or nutrient management.
Reviewer 2 Report
Comments and Suggestions for Authors
1. The study reveals significant inconsistencies in correlations between remote sensing indices and grape quality metrics (phenols, anthocyanins). This undermines the reliability of the proposed approach.
Specific Concern: For example, the lack of consistent relationships between NDVI and phenolic compounds across sites and years weakens the conclusions about the utility of NDVI in predicting berry composition.
2. While some findings suggest the potential for remote sensing, the manuscript generalizes its application without adequately addressing the limitations.
Example: The assertion that "remote sensing thermal emission data could be a key determinant of pH level" is not well-supported given the limited spatial and temporal consistency.
3. Why were certain vegetation indices (CI green, GNDVI) highlighted as more reliable? The paper lacks a discussion on why these indices outperformed others.
4. The PCA analysis accounts for only 44–58% of the data variance, leaving a significant portion unexplained. The authors should discuss how this impacts the robustness of their findings.
5. The spatial maps provided are visually detailed but lack thorough interpretation. The discussion does not adequately address how observed clustering patterns can translate into actionable vineyard management strategies.
6. Would the findings of this study apply to grape varieties other than Cabernet franc? Why or why not?
7. Did the study evaluate the cost-effectiveness of using RPAS compared to traditional sampling methods?
8. The manuscript mentions that the data are unavailable due to their nature. Can the authors clarify this restriction and explore options for partial data sharing to enhance reproducibility?
My suggestion:
9. Could you expand the discussion on why correlations vary significantly across sites and years, focusing on environmental variability, vineyard management practices, and measurement limitations? You could highlight the practical implications of these inconsistencies for large-scale adoption of remote sensing technologies.
10. Additional statistical validation, such as multivariate regression, must be performed to understand better key factors contributing to variability. The authors could simplify data representation using heatmaps or summary tables and include confidence intervals to improve interpretability.
11. Provide actionable examples of how the findings, such as relationships between thermal data and pH, can be implemented for zonal or selective harvesting in real-world vineyard management.
12. Justify why certain indices (CI green, GNDVI) were more reliable, explain the emphasis on 2016 data over 2015, and discuss the potential of other remote sensing platforms or indices to improve accuracy.
13. While remote sensing shows promise, its application is context-dependent and requires further refinement. Explicitly connect the findings to their practical utility and limitations.
Round 2
Reviewer 2 Report
Comments and Suggestions for Authors
All of my questions is already been answered